# Scaffold–Substituent Differentiated Diffusion for Hierarchical Molecule Generation

## Abstract

Deep generative models have emerged as powerful tools for efficiently navigating the vast chemical space and generating molecules with desirable properties. However, existing approaches—particularly diffusion-based models—struggle to effectively model the hierarchical structure of drug-like molecules, which typically consist of a core scaffold and attached substituent functional groups. This hierarchical decomposition is central to modern drug design strategies, where scaffold hopping and lead optimization are applied iteratively to refine molecular structure. While traditional methods can optimize each component separately, they often rely on rule-based heuristics and lack the capacity for joint optimization. To address these limitations, we propose the Scaffold–Substituent Hierarchical Diffusion Model ($S^2$-HDM). It unifies the principles of scaffold hopping and lead optimization within a single generative framework by introducing a differentiated noise schedule for scaffold and substituent atoms. Unlike traditional approaches, $S^2$-HDM implicitly learns the scaffold and substituent hierarchy without pre-defined functional groups, enabling an end-to-end generation pipeline. We validate the effectiveness of our method through extensive experiments, where $S^2$-HDM achieves outstanding performance in multiple generation benchmarks. These results underscore the model's potential to advance drug design by balancing scaffold integrity with substituent diversity, aligning closely with structure-based design principles. The code can be found at https://anonymous.4open.science/r/S2-HDM-6F23.

## 1 INTRODUCTION

The chemical space of drug-like molecules is estimated to contain over $10^{60}$ compounds (Lipinski et al., 1997), making exhaustive exploration infeasible. Recent breakthroughs in machine learning—ranging from image synthesis (Rombach et al., 2022; Dhariwal & Nichol, 2021) to natural language generation (Ouyang et al., 2022)—have motivated the application of generative models to molecular design. Notably, AlphaFold (Jumper et al., 2021; Abramson et al., 2024) has demonstrated the transformative potential of learning-based methods in biochemistry. These successes have inspired growing interest in generative modeling for molecule discovery. In this context, generative models aim to efficiently sample from the underlying distribution of biochemically valid molecules, ensuring that generated compounds exhibit plausible chemical structures and desirable properties.

Impressive progress has recently been made in generative AI for molecular design, with models often categorized based on how they represent and generate molecular structures. For example, (Kusner et al., 2017) (Dai et al., 2018) (Liao et al., 2023) used Variational Autoencoder (Kingma, 2013) to generate molecules represented as 1D SMILES strings. For 2D graph based models (Jin et al., 2018)(Shi et al., 2020)(Luo & Ji, 2022)(Tan et al., 2023)(Kuznetsov & Polykovskiy, 2021)(Vignac et al., 2023)(Kong et al., 2023), these works primarily construct molecular graphs using atom types and bond types, while some adopt structural motifs as the fundamental building blocks. Most of them generate molecules in an autoregressive manner by sequentially adding atoms and bonds. Overall, since molecules exist in 3D space by nature, representing molecules as 3D conformers offers a more holistic depiction to capture both geometric and chemical properties effectively. Among various generative approaches, diffusion models (Hoogeboom et al., 2022; Xu et al., 2022) have emerged as a powerful framework for 3D molecular design. These models generate molecules by gradually adding Gaussian noise to transform structured molecule into a noisy point cloud, and then learning to reverse this process step-by-step to recover realistic molecular conformations. Compared to the

1D- and 2D-based models, these diffusion models offer greater flexibility in capturing geometric constraints, making them well-suited for generating physically plausible 3D molecular structures.

Despite the success of deep generative models in molecular design, most existing approaches—particularly 3D diffusion-based models—fail to capture the hierarchical organization inherent in drug-like molecules. These molecules often exhibit a core–peripheral structure, where the scaffold serves as the central framework defining the overall molecular geometry and pharmacophore orientation, while substituents (or R-groups) are peripheral functional groups that can be strategically modified to influence biological properties (Welsch et al., 2010). This structural decomposition supports modular design strategies, enabling the creation of diverse compound libraries by varying substituents on a fixed scaffold (Hu et al., 2017; Sun et al., 2012), or exploring new core structures through scaffold hopping (Böhm et al., 2004). However, conventional diffusion models treat molecules as undifferentiated collections of atoms, applying uniform generative rules without regard to their structural roles. This lack of structural awareness can limit their effectiveness in generating hierarchical structure, where scaffold integrity and substituent variability are both critical.

A variety of separated design strategies have been proposed, broadly categorized into non-generative and deep generative approaches. Traditional methods such as scaffold hopping and lead optimization rely on predefined similarity metrics to search fragment databases and substitute problematic scaffolds or substituents (Sun et al., 2012; Hessler & Baringhaus, 2010). While effective for local improvements, these non-generative methods are limited in their ability to explore truly novel chemical structures and become inefficient in the vast molecule databases. In contrast, deep generative models have been developed to design either the scaffold or the substituent while holding the other component fixed (Liao et al., 2023; Schneuing et al., 2024). Although this conditional formulation enables modular control, it restricts the generative flexibility by decoupling the structural interdependence between the core and peripheral parts. As a result, such models may overlook globally optimal solutions that require joint optimization for functional and structural compatibility. These limitations give rise to an important research question: How can we develop an effective generative framework that incorporates prior knowledge of scaffold–substituent hierarchy while enabling end-to-end joint optimization of the hierarchical structure for *de novo* molecule design?

To bridge this research gap, as illustrated in Fig. 1, we introduce a Scaffold–Substituent Hierarchical Diffusion Model ($S^2$-HDM), which incorporates a differentiated noise schedule for core and peripheral atoms. The central idea is to encode the molecular hierarchy directly into the generative process: during the forward diffusion process, scaffold atoms are corrupted with less noise than substituent atoms, preserving their structural integrity. In the reverse process, scaffold atoms are denoised earlier and more reliably, establishing a stable core context to guide the subsequent generation of substituents. To support this role-aware generation, we introduce a $S^2$ classifier that dynamically predicts the structural role of each atom throughout the denoising process. Our main contributions are:

- We present $S^2$-HDM, a diffusion-based generative framework inspired by traditional drug design strategies of lead optimization and scaffold hopping. Rather than treating these as separate processes, $S^2$-HDM integrates their objectives into a unified generative model by introducing differentiated noise scheduling for the scaffold and substituent atoms. This hierarchical approach prioritizes the stability of generative scaffold and enables flexible exploration of peripheral functional groups, closely aligning with modular drug design.

- Compared with the vanilla diffusion models and separated design strategies, our model is trained in an end-to-end manner to achieve the hierarchical and joint optimization of scaffold and substituent. Instead of relying on expert-defined functional groups, our model learns the hierarchical structures implicitly, allowing for greater potential for designing novel molecules containing scaffold integrity and substituent variability.

- Extensive experiments on QM9 and GEOM-Drugs datasets demonstrate that our model outperforms baseline methods in molecular stability (by $0.8\%$), uniqueness (by $3.8\%$) and validity (by $2.2\%$). Through empirical observations and ablation studies, we further validate the rationality and effectiveness of each architectural component.

## 2 RELATED WORK

**Diffusion based Molecule Generation.** Diffusion models have recently emerged as powerful frameworks for 3D molecular generation. Early works such as GeoDiff (Xu et al., 2022) and

EDM (Hoogeboom et al., 2022) introduced E(3)-equivariant diffusion processes for modeling conformations and jointly denoising atomic coordinates and types. More recent latent-space approaches like GeoLDM (Xu et al., 2023) and GCDM (Morehead & Cheng, 2024) improve scalability and geometric completeness, enabling larger and more stable structures. Meanwhile, graph-based methods such as DiGress (Vignac et al., 2023) apply discrete diffusion directly on molecular graphs to support scalable and property-conditioned generation. Beyond unconditional generation, conditioned molecular design has seen notable progress. DiffSBDD (Schneuing et al., 2024) conditions on protein binding pockets for structure-based drug design, while D3FG (Lin et al., 2023) models molecules at the functional group level to improve 3D realism and synthetic feasibility.

Research gap of existing methods: Existing diffusion-based models either fail to account for the scaffold-substituent memberships of atoms within a molecule or rely on a pre-defined set of functional groups. This will limit their ability to generate meaningful hierarchical structure for the drug-like small molecules, which are often optimized iteratively by scaffold hopping and lead optimization in the realistic design pipeline. In addition, the pre-defined building blocks constrain the generative diversity, the generative process treats fragments as rigid-body tokens. In contrast, our model incorporates the distinct functional roles of scaffold and substituent atoms without requiring any pre-defined group set, making it more flexible and data-driven.

**Scaffold and Substituent Design.**    Scaffold and substituent manipulation plays a central role in both traditional medicinal chemistry and modern AI-driven molecular design. Recent studies summarize how modifying core scaffolds or peripheral R-groups can lead to enhanced potency, selectivity, and multi-target engagement in small-molecule drug discovery (Acharya et al., 2024). In contrast, modern deep learning approaches explicitly model scaffold and substituent separation to enable controllable molecular generation. (Li et al., 2019) proposed an autoencoder-based model, DeepScaffold, which performs one-shot generation of substituent atoms and bonds conditioned on a given scaffold. (Lim et al., 2020) introduced a graph-based generative model that incrementally constructs molecules by sequentially adding atoms and bonds to a predefined scaffold. (Hu et al., 2023) proposed ScaffoldGVAE, a variational autoencoder disentangling scaffold and substituent representations for scaffold hopping. (Liao et al., 2023) introduced Sc2Mol, a two-step VAE-Transformer framework for scaffold-constrained molecule synthesis. Prompt-based methods such as PromptSMILES (Thomas et al., 2024) and fragment-based models like SAFE (Noutahi et al., 2024) and FragGPT (Yue et al., 2024) further advance scaffold-controlled generation using pretrained chemical language models.

Research gap of existing methods: However, these approaches either require a pre-defined scaffold to generate the substituent—making de novo design infeasible—or explicitly decompose the generation into two separate steps for scaffold and substituent, resulting in increased model complexity and preventing end-to-end training. In contrast, our method enables simultaneous de novo generation of both scaffold and substituent in an end-to-end manner.

## 3 BACKGROUND

**Problem Definition.**    Considering a molecule consisted of $N$ atoms, it can be represented as point clouds in 3D space: $\mathcal{G} = \langle \mathbf{x}, \mathbf{h} \rangle$, where $\mathbf{x} \in \mathbb{R}^{N \times 3}$ is position tensor of every atom, $\mathbf{h} \in \mathbb{R}^{N \times d}$ is node features (e.g. atom types and charges), and $d$ is the dimension of atom features. We focus on two types of generation tasks. i) *Unconditional generation* is defined to model the distribution of training molecules, i.e., $q(\mathcal{G})$, with parameterized neural networks $p_\theta(\mathcal{G})$ and sample from that learned distribution, where $\theta$ is model weights. ii) *Conditional generation* targets at learning the distribution of training molecules as well as their properties $c$, which is denoted by $q(\mathcal{G}, c)$. The parameterized generator $p_\theta(\mathcal{G}|c)$ samples molecules conditioned on the given property $c$.

**Diffusion Models.**    Diffusion models, inspired by non-equilibrium thermodynamics, was first introduced in Sohl-Dickstein et al. (2015) and further advanced in Ho et al. (2020). Specifically, the diffusion models generate data by first gradually adding Gaussian noise to transform a molecule into pure noise through a forward diffusion process, and then learning to reverse this corruption step-by-step via a denoising process to recover realistic data. i) Forward process: Analogous to the process of ink diffusing in water—where the ink gradually disperses and the mixture becomes uniformly cloudy—diffusion models corrupt a molecule by progressively adding noise over a sequence of timesteps. Formally, this corruption is modeled as a forward diffusion process defined by a Markov chain $\mathbf{x}_0, \mathbf{x}_1, \ldots, \mathbf{x}_T$, where $\mathbf{x}_0$ is initiated by the original molecule, such as the 3D position tensor

$\mathbf{x}$, and $\mathbf{x}_T$ is nearly pure noise of point cloud:

$$q(\mathbf{x}_t|\mathbf{x}_{t-1}) = \mathcal{N}\left(\mathbf{x}_t; \sqrt{\alpha_t}\mathbf{x}_{t-1}, \beta_t\mathbf{I}\right), \tag{1}$$

where $\alpha_t$ and $\beta_t$ are pre-scheduled weights to determine the proportion of signal and noise injected at each time step $t$. These parameters are typically chosen such that the final distribution $q(\mathbf{x}_T)$ approximates a standard Gaussian $\mathcal{N}(\mathbf{0}, \mathbf{I})$. A common setting enforces $\alpha_t + \beta_t = 1$ to ensure that the variance of $\mathbf{x}_t$ remains a constant, i.e., identity matrix $\mathbf{I}$. Using this setup, we can easily have: $q(\mathbf{x}_t|\mathbf{x}_0) = \mathcal{N}(\sqrt{\bar{\alpha}_t}\mathbf{x}_0, (1 - \bar{\alpha}_t)\mathbf{I})$ with $\bar{\alpha}_t = \prod_{i=1}^{i=t} \alpha_i$. This forward process is fully specified by fixed hyperparameters and contains no learnable components.

ii) Reverse process: The generative goal of diffusion models is to reverse this corruption process by learning to sample molecules from the reverse conditional distributions. This reverse process approximates the posterior $q(\mathbf{x}_{t-1}|\mathbf{x}_t)$ with a parameterized distribution:

$$p_\theta(\mathbf{x}_{t-1}|\mathbf{x}_t) = \mathcal{N}\left(\mathbf{x}_{t-1}; \boldsymbol{\mu}_\theta(\mathbf{x}_t, t), \rho_t^2\mathbf{I}\right), \boldsymbol{\mu}_\theta(\mathbf{x}_t, t) = \frac{1}{\sqrt{\alpha_t}}\left(\mathbf{x}_t - \frac{1-\alpha_t}{\sqrt{1-\bar{\alpha}_t}}\boldsymbol{\epsilon}_\theta(\mathbf{x}_t, t)\right). \tag{2}$$

$\boldsymbol{\mu}_\theta(\mathbf{x}_t, t)$ is a neural network predicting the mean, and $\rho_t$ is a time-dependent variance, which is often analytically derived as: $\rho_t = \frac{\beta_t(1-\sqrt{\bar{\alpha}_{t-1}})}{(1-\bar{\alpha}_t)}$. $\boldsymbol{\epsilon}_\theta$ predicts the noise component added at each step.

The optimization of generative model used in the reverse process is to maximize the likelihood of training molecules, i.e., $p_\theta(\mathbf{x})$. Due to the difficulty of accessing the real $p_\theta(\mathbf{x})$, training is instead performed by maximizing a variational bound, specifically the Evidence Lower BOund (ELBO), given by: $\text{ELBO} = \mathbb{E}_{q(\mathbf{x}_{1:T}|\mathbf{x}_0)}[\log \frac{q(\mathbf{x}_T|\mathbf{x}_0)}{p_\theta(\mathbf{x}_T)} + \sum_{t=2}^T \log \frac{q(\mathbf{x}_{t-1}|\mathbf{x}_t,\mathbf{x}_0)}{p_\theta(\mathbf{x}_{t-1}|\mathbf{x}_t)} - \log p_\theta(\mathbf{x}_0|\mathbf{x}_1)]$. Under this framework, the training objective can be simplified to a denoising score matching loss:

$$\mathcal{L}_{\text{noise}} = \mathbb{E}_{\mathbf{x}_0, \boldsymbol{\epsilon} \sim \mathcal{N}(\mathbf{0},\mathbf{I}), t \sim \text{Uniform}(0,T)} \left[||\boldsymbol{\epsilon} - \boldsymbol{\epsilon}_\theta(\mathbf{x}_t, t)||^2\right], \tag{3}$$

where $\boldsymbol{\epsilon}$ is the known noise used to construct $\mathbf{x}_t = \sqrt{\bar{\alpha}_t}\mathbf{x}_0 + \sqrt{1-\bar{\alpha}_t}\boldsymbol{\epsilon}$. Over successive reverse steps, the model aims to iteratively remove noise, ultimately reconstructing $\mathbf{x}_0$ such that the learned data distribution $p_\theta(\mathbf{x}_0)$ converges to the true data distribution $q(\mathbf{x}_0)$.

# 4 SCAFFOLD–SUBSTITUENT HIERARCHICAL DIFFUSION MODEL (S$^2$-HDM)

Despite recent progress in diffusion models for molecular design, their denoising-based generative process often overlooks the inherent core–peripheral structure of a molecule—namely, the separation between scaffolds and substituents. Especially in the drug-like molecules, the scaffold typically defines the molecule's core geometry and conformational rigidity, contributing to metabolic stability. In contrast, substituents are functional groups that modulate peripheral interactions and influence properties like solubility. This separation allows researchers to establish a stable core structure and afterward flexibly optimize peripheral properties, which enhances synthetic feasibility. However, existing diffusion models typically treat the molecule as an undifferentiated whole, applying the same noise schedule across all atoms regardless of their structural roles. To overcome this limitation, we propose S$^2$-HDM that applies differentiated noise schedules to scaffolds and substituents, allowing scaffold sketching in early denoising and then enabling flexible exploration of substituent space. As illustrated in Fig. 1, the proposed framework consists of three components: (i) a hierarchical noise scheduler that distinguishes between scaffold and substituent atoms, (ii) a scaffold–substituent classifier that dynamically estimates the structural role of each atom during generation, and (iii) a modified denoising model that updates atomic positions and features based on their roles.

**Hierarchical Noise Scheduler.** The proposed hierarchical mechanism is motivated by two typical types of traditional design approaches for drug-like small molecules: scaffold hopping (Acharya et al., 2024) and lead optimization (Barcelos et al., 2022). Scaffold hopping aims at discovering new compounds with similar biological activity by altering the molecular scaffold while keeping the key functional groups of substituent intact. Lead optimization aims to improve the drug-like properties of a lead compound by systematically adding, removing, or modifying its substituent, i.e., the functional groups attached to the core scaffold. These two approaches are complementary and form a cornerstone of modern drug discovery pipelines.

Inspired by this two-stage design process, we propose to integrate its principles into the diffusion model by introducing differentiated noise schedules for scaffold and substituent atoms. Specifically,

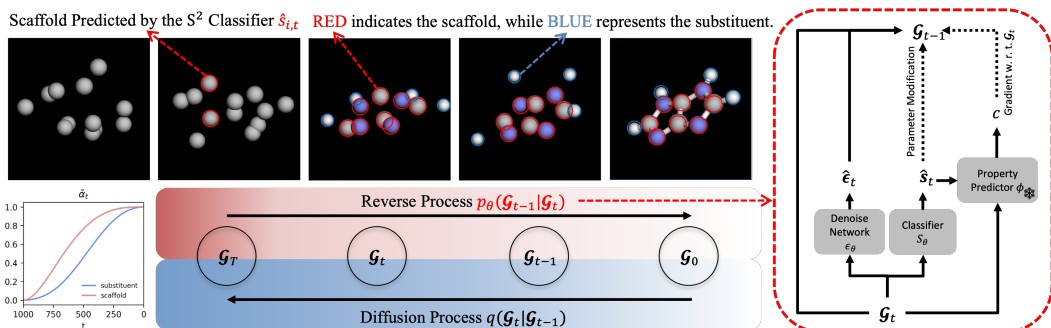

Figure 1: Illustration of S$^2$-HDM. Atoms marked with red circle are predicted to be scaffold and prioritized in denoising process to provide stable context. Atoms marked with blue circle are predicted to be substituent and denoised mainly in later stage. At each timestep $t$, $\mathcal{G}_t$ is fed into the EGNN to produce both the predicted noise $\hat{\epsilon}_t$ and the predicted scaffold label $\hat{s}_t$. Simultaneously, $\mathcal{G}_t$ is input into the property predictor to compute guidance gradient. The property predictor is trained beforehand and kept frozen during the denoising process. The predicted scaffold label $\hat{s}_t$ is then used to adaptively update position and feature tensors of scaffold and substituent atoms in reverse process.

for a molecule $\mathcal{G} = \langle \mathbf{x}, \mathbf{h} \rangle$, we identify the scaffold–substituent structure using computational chemistry tools such as RDKit[1], yielding a tag vector $\mathbf{s} = (s_1, \ldots, s_N) \in \{0,1\}^N$, where $s_i = 1$ denotes a scaffold atom and $s_i = 0$ indicates a substituent atom. To encode structural hierarchy, we assign distinct weighting factors $\bar{\alpha}_t$ defined in Eq.(1) based on these tags. Scaffold atoms receive lower noise levels during the forward diffusion process, preserving structural integrity, while substituent atoms receive higher noise, encouraging flexible exploration. Accordingly, during the reverse generative process, scaffold atoms are denoised earlier, providing a stable core context for subsequent generation of substituents. Formally, let $\bar{\alpha}_{sc,t} = \bar{\alpha}_t$ denote the weighting factor for scaffold atoms. The weighting factor for substituent atoms is modulated as:

$$\bar{\alpha}_{su,t} = \bar{\alpha}_t \omega_t, \quad \omega_t = \cos\left(0.5\pi t/T\right)^{\eta_0} * 0.8 + 0.2. \tag{4}$$

$\eta_0$ is the hyper-parameter that controls the ratio between the weighting factor of substituent and scaffold atoms. The scheduled values of $\bar{\alpha}_{sc,t}$ and $\bar{\alpha}_{su,t}$ w.r.t. time steps are shown in Fig. 1. The scaffold requires faster denoising, meaning its $\bar{\alpha}_{sc,t}$ remains closer to 1 over fewer denoising steps, whereas the substituent requires a slower denoising process. Therefore, we enforce $\bar{\alpha}_{sc,t} > \bar{\alpha}_{su,t}$ at all timesteps, with both schedules satisfying the boundary conditions: $\bar{\alpha}_{sc,t} = 1, \bar{\alpha}_{su,t} = 1$ at step 0 and $\bar{\alpha}_{sc,t} = 0, \bar{\alpha}_{su,t} = 0$ at step $T$. More details are discussed in Appendix. A.1.

Let $\mathcal{G}_t = \langle \mathbf{x}_t, \mathbf{h}_t \rangle$ denotes the noisy molecule obtained at time step $t$, where $\mathbf{x}_t$ and $\mathbf{h}_t$ are corrupted position and feature tensors by adding Gaussian noise $\langle \epsilon_t^{(\mathbf{x})}, \epsilon_t^{(\mathbf{h})} \rangle$ on input molecule $\mathcal{G}$. According to the forward process defined in Eq.(1), the corrupted tensors are obtained by:

$$\begin{cases} \mathbf{x}_t = (\mathbf{s} \cdot \sqrt{\bar{\alpha}_{sc,t}} + (\mathbf{1} - \mathbf{s}) \cdot \sqrt{\bar{\alpha}_{su,t}})\mathbf{x}_0 + (\mathbf{s} \cdot \sqrt{1 - \bar{\alpha}_{sc,t}} + (\mathbf{1} - \mathbf{s}) \cdot \sqrt{1 - \bar{\alpha}_{su,t}})\epsilon_t^{(\mathbf{x})}, \\ \mathbf{h}_t = (\mathbf{s} \cdot \sqrt{\bar{\alpha}_{sc,t}} + (\mathbf{1} - \mathbf{s}) \cdot \sqrt{\bar{\alpha}_{su,t}})\mathbf{h}_0 + (\mathbf{s} \cdot \sqrt{1 - \bar{\alpha}_{sc,t}} + (\mathbf{1} - \mathbf{s}) \cdot \sqrt{1 - \bar{\alpha}_{su,t}})\epsilon_t^{(\mathbf{h})} \end{cases} \tag{5}$$

**Scaffold–Substituent ($S^2$) Classifier.** Given the use of differentiated noise scheduling in the forward process, a key challenge arises in the reverse process: how to accurately distinguish scaffold and substituent atoms from noisy molecular representations—especially when explicit scaffold–substituent annotations are unavailable at inference time. Addressing this challenge is crucial for enabling the hierarchical generation strategy, where scaffold atoms are denoised earlier to provide structural context for subsequent substituent generation. To this end, we introduce an $S^2$ binary classifier that dynamically predicts the structural role of each atom during denoising. As shown within the red box of Fig. 1, given the noisy input $\mathcal{G}_t$, the classifier $S_\theta(\mathcal{G}_t)$ estimates the scaffold probability for each atom, producing a prediction vector $\hat{s}_t = (\hat{s}_{1,t}, ..., \hat{s}_{N,t})$, where $\hat{s}_{i,t}$ indicates the probability that atom $i$ belongs to the scaffold. Classifier $S_\theta(\mathcal{G}_t)$ is implemented by

---

[1]RDKit: Open-source cheminformatics. https://www.rdkit.org

equivariant graph neural networks Satorras et al. (2021) (EGNN) to learn the 3D structure and produce precise differentiation. The classifier is trained using a standard binary cross-entropy loss: $\mathcal{L}_{\text{CLS}} = \mathbb{E}_{\mathcal{G}, \epsilon, t} \left[ \frac{1}{N} \sum_{i=1}^{N} (s_i \log (\hat{s}_i) + (1 - s_i) \log (1 - \hat{s}_{i,t})) \right]$, where $s_i \in \{0, 1\}$ is the ground-truth scaffold label for atom $i$ obtained above.

**Modified Denoising Model.** According to Eq.(2), the approximated posterior $p_\theta(\mathcal{G}_{t-1}|\mathcal{G}_t)$ in the original diffusion model can be implemented via the following update rule:

$$\mathcal{G}_{t-1} = \frac{1}{\sqrt{\alpha_t}} \left( \mathcal{G}_t - \frac{1 - \alpha_t}{\sqrt{1 - \bar{\alpha}_t}} \epsilon_\theta(\mathcal{G}_t, t) \right) + \rho_t \epsilon, \quad \epsilon \in \mathcal{N}(\mathbf{0}, \mathbf{I}). \tag{6}$$

Noise $\epsilon$ is randomly sampled noise added to the generated positions and features to promote sample diversity, and $\epsilon_\theta(\mathcal{G}_t, t)$ is the predicted noise output conditioned on the noisy molecule and the current timestep. We leverage EGNN as backbone model for $\epsilon_\theta$, which estimates the noise added during the forward process. However, unlike previous work, our denoising process should distinguish between scaffold and substituent atoms, which were perturbed using different noise schedules with weight factors $\bar{\alpha}_{sc,t}$ and $\bar{\alpha}_{su,t}$, respectively. Thus we modify the posterior during denoising as:

$$p_\theta(\mathcal{G}_{i,t-1}|\mathcal{G}_t) = S_\theta(s_{i,t} = 1|\mathcal{G}_t)p_\theta(\mathcal{G}_{i,t-1}|s_{i,t} = 1, \mathcal{G}_t) + S_\theta(s_{i,t} = 0|\mathcal{G}_t)p_\theta(\mathcal{G}_{i,t-1}|s_{i,t} = 0, \mathcal{G}_t). \tag{7}$$

$S_\theta(s_{i,t} = 1|\mathcal{G}_t)$ and $S_\theta(s_{i,t} = 0|\mathcal{G}_t)$ are the predicted scaffold probabilities from the $S^2$ classifier. $p_\theta(\mathcal{G}_{i,t-1}|s_{i,t} = 1, \mathcal{G}_t)$ and $p_\theta(\mathcal{G}_{i,t-1}|s_{i,t} = 0, \mathcal{G}_t)$ are obtained by replacing $\bar{\alpha}_t$ of Eq.(6) with $\bar{\alpha}_{sc,t}$ and $\bar{\alpha}_{su,t}$, respectively. The same substitution applies for other constants like $\alpha_t$ and $\rho_t$. Please refer to the Appendix A.2 for details and Appendix.A.3 for the proof of Eq.(7) . This formulation enables soft role-aware denoising, where atoms are updated based on their estimated probabilities of being part of the scaffold or substituent. This aligns the reverse process with the differentiated noise injection applied in Eq.(5). The trainable components of posterior $p_\theta(\mathcal{G}_{t-1}|\mathcal{G}_t)$ include the noise prediction network $\epsilon_\theta$ and the $S^2$ classifier, which are jointly optimized using denoising loss $\mathcal{L}_{\text{noise}}$ and cross-entropy loss $\mathcal{L}_{\text{CLS}}$.

**Model Design Details.** Based on the above framework, we elaborate the additional details of model and training design. First, to enhance the hierarchical structure–property consistency of the generated molecules, we introduce a property-guided refinement mechanism that steers the generation process toward molecules with desirable properties. The experiment result in Tab. 3 justifies that this property-guided denoising process synergizes with the hierarchical generation but can deteriorate the whole-molecule 3D diffusion. Second, since the intermediate molecular representation $\mathcal{G}_t$ is inherently noisy, the scaffold probability output becomes unreliable when $t$ approaches $T$. To address this, we restrict the application stages of both the classifier and the refinement mechanism. Actually, we find that the classification is only relatively stable and accurate when $t < 0.4T$. As shown in our experiment, the classifier can already accurately predict the scaffold/substituent tag of atoms at epoch 20 when $t < 0.4T$, but cannot achieve a high accuracy even after hundreds of epochs when $t \to T$. Similar to what has been mentioned in Han et al. (2023), the same phenomenon occurs with the property predictor in our case. The property-guidance term will not be applied when $t > 0.4T$. Finally, to avoid the wrong hierarchical structure generation and property-guided denoising, we introduce a time-dependent weighting factor and a prediction momentum. Those two methods try to stabilize the predicted scaffold probability by adding a weighted momentum. The weighting factor $\psi_t$ is formulated as $\psi_t = \frac{1}{1 + e^{100(t/T - 0.3)}}$ and the predicted scaffold probability $\hat{s}_t$ is re-formulated as $\psi_t \hat{s}_t + (1 - \psi_t) \hat{s}_{t+1}$. More details can be found in Appendix.A.4.

# 5 EXPERIMENT

**Evaluation Tasks and Datasets.** Following previous practice (Hoogeboom et al., 2022)(Xu et al., 2023)(Cornet et al., 2024) on molecule generation in 3D space, we evaluate our method on two settings: unconditional and conditional generations. The datasets of QM9 (Ramakrishnan et al., 2014) and GEOM-DRUG dataset (Axelrod & Gomez-Bombarelli, 2022) are treated as benchmarks, which have been widely adopted in baseline works. The dataset details are listed in Appendix A.6.

## 5.1 UNCONDITIONAL MOLECULE GENERATION

**Evaluation Metrics.** To evaluate model performance, following the previous work (Hoogeboom et al., 2022), we measure the model's capability to learn the data distribution by calculating the

chemical validity of the generated molecules. With the generated 3D molecule conformers, we first determine the bond type (single, double, triple or none) using the distance between atoms and the atom type. Then, given the bond type and atom type (molecular graph), we calculate the atom stability and the molecule stability of the molecule. Atom stability (A.S.) is the proportion of atoms that have the right valency. Molecule stability (M.S.) is the proportion of the generated molecules of which all the atoms are stable. Also, we report the validity (V) and uniqueness (U) of the generated molecules. Validity is the proportion of the valid molecule measured by RDkit. Uniqueness is the unique molecules among all the generated molecules. The metrics are calculated on 10,000 samples generated from each method. We report the mean and standard deviation from three repeated runs.

**Baselines.** We compare the proposed $S^2$-HDM method with three types of generative models: autoregressive model, flow model and diffusion models. i) Autoregressive model: G-Schnet (Gebauer et al., 2019) is an autoregressive model generating 3d point set that respect the rotational invariance of the targeted structure. ii) Flow model: E-NF (Garcia Satorras et al., 2021) is a normalizing flow based model that take $E(n)$ graph neural network as the invertible equivariant function. Geometric Bayesian Flow Networks (GeoBFN) (Song et al., 2023) incorporates Bayesian inference into the flow model, leading to better efficiency and quality. iii) Diffusion model: Equivariant graph diffusion model (EDM) (Hoogeboom et al., 2022) combines the equivariant graph network with the diffusion model while Graph Diffusion model (GDM) is the non-equivariant variation of EDM. Geometric Latent Diffusion Model (GeoLDM) (Xu et al., 2023) is the first diffusion model that utilize the autoencoder to project the molecule into a latent space and perform the diffusion process in the latent space. Equivariant Neural Diffusion (END) (Cornet et al., 2024) features a learnable forward process instead of a pre-specified one for enhanced generative modeling. Hierarchical Diffusion-based model (HierDiff) (Qiang et al., 2023) uses a two stage coarse-to-fine strategy to generate fragment-level and atom-level structures sequentially.

Table 1: Comparison with key baselines on QM9 and GEOM-Drugs datasets. **The higher is the better**. A./M. S.: Atom or molecule stability. V&U/S: Valid and unique/atom stable.

| Category | Methods | QM9 | | | | GEOM-Drugs | | |
|---|---|---|---|---|---|---|---|---|
| | | A. S. (%) | M. S. (%) | V (%) | V&U (%) | A. S. (%) | V (%) | V&S (%) |
| Autoregressive | G-Schnet | 95.7 | 68.1 | 85.5 | 80.3 | - | - | - |
| Flow | E-NF | 85.0 | 4.9 | 40.2 | 39.4 | - | - | - |
| | GeoBFN | 99.1±0.1 | 90.9±0.2 | 95.3±0.1 | 93.0±0.1 | 85.6 | 92.1 | 78.8 |
| Diffusion | GDM | 97.0 | 63.2 | - | - | 75.0 | 90.8 | 68.1 |
| | GDM-aug | 97.6 | 71.6 | 90.4 | 89.5 | 77.7 | 91.8 | 71.3 |
| | GeoLDM | 98.9±0.1 | 89.4±0.5 | 93.8±0.4 | 92.7±0.5 | 84.4 | **99.3** | 83.8 |
| | EDM | 98.7±0.1 | 82.0±0.4 | 91.9±0.6 | 90.7±0.6 | 81.3 | 92.6 | 75.3 |
| | END | 98.9±0.0 | 89.1±0.1 | 94.8±0.1 | 92.6±0.2 | **87.0** | 89.2 | 77.6 |
| Hierarchical Diffusion | HierDiff-E | - | - | 87.8 | 86.0 | - | 94.0 | - |
| | HierDiff-P | - | - | 83.6 | 82.3 | - | 90.4 | - |
| | $S^2$-HDM | **99.2**±0.1 | **91.7**±0.4 | **97.5**±0.3 | **96.8**±0.3 | 85.4 | 98.4 | **84.0** |

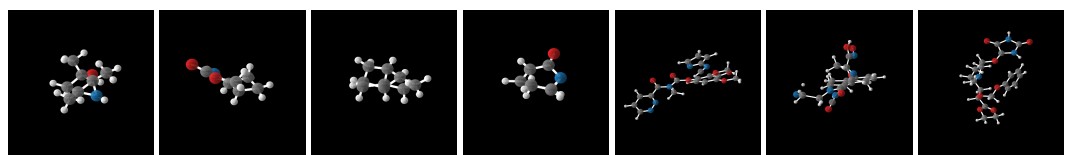

Figure 2: Samples generated by $S^2$-HDM trained on QM9 (left four) and GEOM-Drugs (right three).

**Obs. 1: Our hierarchical diffusion model delivers superior generation results compared with the baseline models on unconditional molecule generation.** Results of both our method and the baseline methods are reported in the Tab. 1. Our method surpasses the baseline methods in terms of Valid (+2.2%), Molecule stable (+0.8%) and Valid&Unique (+3.8%) on QM9 and Valid&Stable (+0.2%) on GEOM-Drugs. The observed performance gain can be attributed to the staged generation of scaffold followed by sustituent, which reflects established best practices in medicinal chemistry and molecular design. We showcase a selection of molecule samples generated from the QM9 and GEOM-Drugs datasets in Fig. 2. More experiment details can be found in Appendix A.5.

**Obs. 2: The scaffold and sustituent are indeed generated and subsequently refined at distinct stages of the denoising process.** We visualized the molecular structures at different stages of the generation process in Fig. 3. Intuitively, we observe that scaffold atoms are indeed generated earlier than sustituent atoms, and they remain relatively stable during subsequent timesteps. More specifically,

the positions of scaffold atoms stabilize relatively early in the generation process, whereas their atom types, which depend on the surrounding atomic context, are determined only at the final stages. In contrast, both the positions and atom types of sustituent atoms tend to stabilize at later timesteps. Fig. 4 shows the progression of atomic displacements across denoising timesteps, providing empirical support for the aforementioned observation.

Reverse Process $p_\theta(\mathcal{G}_{t-1}|\mathcal{G}_t)$

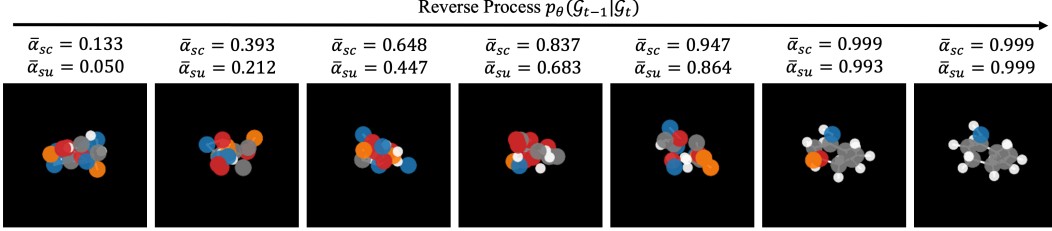

Figure 3: Visualization of the $S^2$-HDM denoising process, where scaffold atoms are denoised earlier.

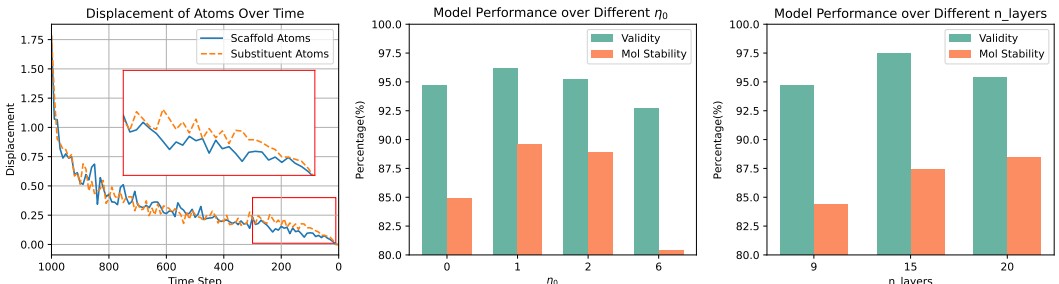

Figure 4: **Left:** Displacement of the scaffold/sustituent atoms over the denoising steps. **Middle&Right:** $S^2$-HDM's performance over a range of noise difference ratios $\eta_0$ and model layers.

**Obs. 3: Ablation studies reveal that different components of our framework play complementary roles in enhancing performance.** We performed a series of controlled ablation studies to systematically evaluate the

Table 2: Ablation study on QM9.

| Hierarchical Scheduler and Classifier | Modified Denoising Model | Property-guided Refinement | QM9 Mol Stable (%) | QM9 Valid (%) |
|---|---|---|---|---|
| ✓ | ✓ | ✓ | 91.7±0.4 | 97.5±0.3 |
| ✓ | ✓ | | 89.6±0.4 | 95.9±0.5 |
| ✓ | | | 86.6±0.3 | 94.0±0.4 |
| | | | 82.0±0.4 | 91.9±0.6 |

impact of each design component. The hierarchical noise scheduler and the scaffold/sustituent classifier modules are deeply coupled in the model design. Therefore, in the ablation study, they are treated as a single unit and are not evaluated separately. As shown in Tab. 2, the design that achieved the largest performance gain is the combination of the hierarchical noise scheduler and the scaffold/sustituent classifier, leading to a $4.6\%$ improvement in molecule stability and a $2.1\%$ improvement in validity compared with the ablated variant on these two modules. A possible explanation is that the training of the classifier incorporates additional scaffold/sustituent labels generated by RDKit, providing the model with extra supervision signals that facilitate better discrimination between different structural components. By progressively refining the scaffold and sustituent in separate stages, the model benefits from a more stable contextual structure, thereby promoting the generation of chemically more plausible molecules. In addition, the inclusion of modified denoising enhance the stability of generated molecules, justifying the denoising rationality to generate hierarchical structure. The property-guided refinement uses classifier-derived gradient guidance in denoising to enhance the consistency of generated hierarchy with molecular energy levels, thus further improving performance.

**Obs. 4: $S^2$-HDM achieves the optimal performance with modest network depth and noise difference ratio.** We primarily investigated the impact of the network depth and the parameter $\eta_0$, which controls the noise difference between scaffold and sustituent atoms, on model performance. The results are shown in the Fig. 4. We compared models with 9, 15, and 20 layers, and the experiments suggest that the 9-layer baseline model is not the optimal choice; as the depth increases, model performance continues to improve. Regarding the effect of $\eta_0$, applying a differentiated noise schedule yields better results than using a uniform noise schedule ($\eta_0 = 0$), and a moderate noise difference ($\eta_0 \in [1, 2]$) outperforms a large noise difference ($\eta_0 = 6$).

**Obs. 5: $S^2$-HDM demonstrates clear advantages in both training and inference efficiency.** As illustrated in Fig. 5, despite the hierarchical design, our method converges significantly faster

than EDM to reach comparable quality metrics of Validity and Molecular Stability: To reach 80% validity, $S^2$-HDM required 46 epochs, 14 hours, while EDM took 123 epochs, 31 hours. To reach 80% Mol Stability, $S^2$-HDM required 208 epochs, 63 hours, while EDM took 631 epochs, 159 hours. Compared with the vanilla whole-molecule diffusion, the hierarchical generation of scaffold and substituent atoms prones to formulate effective molecules.

To fairly compare the trade-off between inference cost and model performance, we compare model parameters, sampling time, and performance of EDM augmented with property guidance in Tab. 3. First, while $S^2$-HDM introduces modest overhead in sampling time ($\sim 1.1\times$ compared to EDM), this is a necessary trade-off for improved generation quality and controllability. According to PTST metric, $S^2$-HDM achieves higher validity of sampling molecules per second, while it only pay negligible time cost based on metric STPR. Second, the inclusion of property guidance on EDM yields poor results compared the EDM-only performance in Tab. 1. This justifies the design rationale of coupling hierarchical generation with property-classifier gradient guidance.

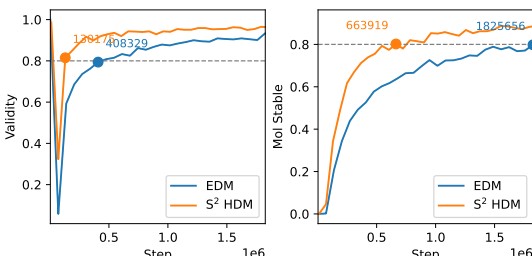

Figure 5: Model performance v.s. training steps.

Table 3: Comparison of inference efficiency metrics between EDM (with property guidance) and $S^2$-HDM. Sampling-Time-to-Parameter Ratio **(STPR)**: Reflects how computational-efficient the model is during inference. Performance-to-Sampling-Time Ratio **(PTST)**: Captures how many stable molecules each second of sampling yields.

| Method | Parameters | Sampling Time (s / 10k samples)↓ | V↑ | A. S.↑ | M. S.↑ | STPR↓ | PTST↑ |
|---|---|---|---|---|---|---|---|
| EDM + Property Guidance | 3M | **65** | 92.1 | 98.9 | 84.3 | 21.7 | **1.29** |
| $S^2$-HDM | 6M | 72 | **97.5** | **99.2** | **91.7** | **12.0** | 1.27 |

## 5.2 CONDITIONAL MOLECULE GENERATION

**Evaluation Metrics and Baselines.** We evaluate the Mean Absolute Error (MAE) of the given property and the property of the generated molecules, which is predicted by the property prediction network (Hoogeboom et al., 2022). The properties we use include polarizability $\alpha$, highest occupied molecular orbital energy $\varepsilon_{\text{HOMO}}$, lowest unoccupied molecular orbital energy $\varepsilon_{\text{LUMO}}$, HOMO-LUMO gap $\Delta\varepsilon$, dipole moment $\mu$ and heat capacity $C_v$. More details are in Appendix A.5. Following (Hoogeboom et al., 2022) and (Xu et al., 2023), we list two additional baselines named $Random$ and $N_{\text{atoms}}$. $Random$ shuffles the property labels and represents the upper bound of the MAE. $N_{\text{atoms}}$ take only the number of atoms as the input.

Table 4: MAE of conditional 3D molecule generation on QM9. **The lower is the better**.

| **Property** | $\alpha$ | $\Delta\varepsilon$ | $\varepsilon_{\text{HOMO}}$ | $\varepsilon_{\text{LUMO}}$ | $\mu$ | $C_v$ |
|---|---|---|---|---|---|---|
| **Units** | Bohr$^3$ | meV | meV | meV | D | $\frac{\text{cal}}{\text{mol K}}$ |
| $Random$ | 9.01 | 1470 | 645 | 1457 | 1.616 | 6.857 |
| $N_{\text{atoms}}$ | 3.86 | 866 | 426 | 813 | 1.053 | 1.971 |
| EDM | 2.76 | 655 | 356 | 584 | 1.111 | 1.101 |
| GeoLDM | 2.37 | 587 | 340 | 522 | 1.108 | 1.025 |
| GeoBFN | 2.34 | 577 | 328 | 516 | 0.998 | 0.949 |
| **$S^2$-HDM** | **1.34** | **465** | **242** | **417** | **0.873** | **0.924** |

**Results and Analysis.** As shown in Tab. 4, our method exceeds the baseline model in all quantum properties, with MAE decreasing on average by $20.5\%$. The comparison between $N_{\text{atoms}}$ and $S^2$-HDM shows that the proposed model is capable of effectively embedding property information into the molecular generation process.

## 6 CONCLUSION

In this work, we present $S^2$-HDM, a hierarchical diffusion model with a differentiated noise schedule, which effectively integrates domain knowledge from traditional medicinal chemistry, i.e., scaffold hopping and lead optimization, into a fully end-to-end generative framework. Unlike the separated design strategies that rely on two-stage iterative refinement and pre-defined functional groups, $S^2$-HDM introduces a differentiable approach that implicitly learns the scaffold-substituent hierarchy and generates molecules by prioritizing scaffold establishment and then enabling flexible substituent exploration. Our model demonstrates consistent improvements over baseline methods in both stability and validity on standard benchmarks, including QM9 and GEOM-Drugs. Through comprehensive ablation studies and qualitative analysis, we validate the design of each component and show that $S^2$-HDM is capable of producing structurally diverse and chemically meaningful molecules.

## ETHICS STATEMENT

This study develops a hierarchical diffusion method using only publicly available molecule datasets and publicly released foundation models, in accordance with the ICLR Code of Ethics. We did not collect new data, involve human subjects, or access protected health information.

## REPRODUCIBILITY STATEMENT

We ensure reproducibility by documenting implementation details in method section, including model configuration and optimization. Dataset descriptions and evaluation metrics are detailed in experiment section. An anonymous GitHub repository link provides code, configs, and scripts to reproduce molecule generation results. We fix random seeds and report mean and standard variance performance from multiple independent runs.

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

# A APPENDIX

## A.1 DETAILS OF $\omega_t$

Our design is motivated by three main considerations:

- At $t = 0$, we set $\omega_t = 1$;
- For $t > 0$, we ensure $\omega_t \leq 1$;
- At $t = T$, for numerical stability, $\omega_t$ should be greater than 0 and not too small.

In Figure. 4, we conducted an ablation study to select the most suitable $\eta_0$. We report the performance of different implementations under the same limited training epochs (which are less than 1300) as follows:

Table 5: Ablation study on the choice of $\omega_t$. The current parameter choice yields the fastest convergence and best performance.

| $\omega_t$ | Valid(%) | Atom Stable(%) | Mol Stable(%) | Comment |
|---|---|---|---|---|
| $\cos(0.5\pi t/T) \times 0.8 + 0.2$ | 92.7 | 98.4 | 80.2 | |
| $\cos(0.5\pi t/T)$ | NA | NA | NA | Training fails due to numerical instability. |
| $0.8(1 - t/T) + 0.2$ | 87.6 | 97.7 | 73.2 | |
| $0.8(1 - t/T)^2 + 0.2$ | 88.2 | 97.8 | 74.5 | |
| $0.8(1 - t/T)^6 + 0.2$ | 59.2 | 91.9 | 41.5 | |
| $0.99(1 - t/T)^2 + 0.01$ | 83.9 | 96.4 | 70.4 | |

## A.2 DETAILS OF THE PARAMETER MODIFICATION

Given the weighting factor $\bar{\alpha}_t$, we have:

$$\bar{\alpha}_{sc,t} = \bar{\alpha}_t, \tag{8}$$

$$\alpha_{sc,t} = \begin{cases} \frac{\bar{\alpha}_{sc,t}}{\bar{\alpha}_{sc,t-1}}, & \text{if } i \geq 2 \\ \bar{\alpha}_{sc,1}, & \text{if } i = 1 \end{cases} \tag{9}$$

$$\rho_{sc,t} = \frac{(1 - \alpha_{sc,t})(1 - \sqrt{\bar{\alpha}_{sc,t-1}})}{(1 - \bar{\alpha}_{sc,t})}, \tag{10}$$

$$\bar{\alpha}_{su,t} = \bar{\alpha}_t \omega_t \tag{11}$$

$$\alpha_{su,t} = \begin{cases} \frac{\bar{\alpha}_{su,t}}{\bar{\alpha}_{su,t-1}}, & \text{if } i \geq 2 \\ \bar{\alpha}_{su,1}, & \text{if } i = 1 \end{cases} \tag{12}$$

$$\rho_{su,t} = \frac{(1 - \alpha_{su,t})(1 - \sqrt{\bar{\alpha}_{su,t-1}})}{(1 - \bar{\alpha}_{su,t})}. \tag{13}$$

We let $\bar{\alpha}_t = (1 - 2\tau)(1 - (t/T)^2) + \tau$ with $\tau = 10^{-5}$ to avoid numerically unstable issues.

A.3 PROOF OF EQUATION.(7)

Below we provide a detailed clarification and derivation of Eq.(7) and how ELBO holds.

As shown in the original DDPM formulation(Ho et al., 2020):

$$q(\mathbf{x}_{t-1} \mid \mathbf{x}_t, \mathbf{x}_0) = \mathcal{N}\left(\mathbf{x}_{t-1}; \tilde{\boldsymbol{\mu}}_t(\mathbf{x}_t, \mathbf{x}_0), \tilde{\beta}_t \mathbf{I}\right), \tag{14}$$

and the DDIM formulation(Song et al., 2020):

$$q_\sigma(\mathbf{x}_{t-1} \mid \mathbf{x}_t, \mathbf{x}_0) = \mathcal{N}\left(\sqrt{\bar{\alpha}_{t-1}}\mathbf{x}_0 + \sqrt{1 - \bar{\alpha}_{t-1} - \sigma_t^2} \cdot \frac{\mathbf{x}_t - \sqrt{\bar{\alpha}_t}\mathbf{x}_0}{\sqrt{1 - \bar{\alpha}_t}}, \sigma_t^2 \mathbf{I}\right), \tag{15}$$

the forward posterior always uses a **diagonal covariance matrix**. Any correlation among elements is entirely captured by the **mean vector**.

In the reverse process, the model defines:

$$p_\theta\left(\mathbf{x}_{t-1} \mid \mathbf{x}_t\right), \tag{16}$$

by approximating $\mathbf{x}_0$ in the forward posterior with $\hat{\mathbf{x}}_0(\mathbf{x}_t)$ or $\hat{\epsilon}_0(\mathbf{x}_t)$, which implies that any inter-element dependencies are **introduced solely by the denoising network**.

Therefore, the use of molecule-level symbols such as $s$ and $\mathcal{G}$ can (and should) be replaced by atom-level symbols. For example, instead of:

$$q(\boldsymbol{\mathcal{G}}_t \mid \boldsymbol{\mathcal{G}}_0), \tag{17}$$

we should consider:

$$q(\mathcal{G}_{i,t} \mid \mathcal{G}_{i,0}), \tag{18}$$

where $i$ denotes the $i$-th atom and obviously $q(\mathcal{G}_{i,t} \mid \mathcal{G}_{i,0}) = q(\mathcal{G}_{i,t} \mid \boldsymbol{\mathcal{G}}_0)$ .

Accordingly, the posterior:

$$q(\boldsymbol{\mathcal{G}}_{t-1} \mid \boldsymbol{\mathcal{G}}_t, \boldsymbol{\mathcal{G}}_0), \tag{19}$$

should be refined to the per-atom formulation:

$$q(\mathcal{G}_{i,t-1} \mid \boldsymbol{\mathcal{G}}_t, \boldsymbol{\mathcal{G}}_0) = q(\mathcal{G}_{i,t-1} \mid s_{i,t} = 1, \boldsymbol{\mathcal{G}}_t, \boldsymbol{\mathcal{G}}_0) \cdot q(s_{i,t} = 1 \mid \boldsymbol{\mathcal{G}}_t, \boldsymbol{\mathcal{G}}_0)$$
$$+ q(\mathcal{G}_{i,t-1} \mid s_{i,t} = 0, \boldsymbol{\mathcal{G}}_t, \boldsymbol{\mathcal{G}}_0) \cdot q(s_{i,t} = 0 \mid \boldsymbol{\mathcal{G}}_t, \boldsymbol{\mathcal{G}}_0). \tag{20}$$

As $p_\theta(\boldsymbol{\mathcal{G}}_{t-1} \mid \boldsymbol{\mathcal{G}}_t)$ is trained to minimize the KL divergence with the true posterior $q(\boldsymbol{\mathcal{G}}_{t-1} \mid \boldsymbol{\mathcal{G}}_t, \boldsymbol{\mathcal{G}}_0)$ (i.e., minimize the ELBO), we aim to show that the forward posterior takes the form of Eq.(20), which is a mixture of two Gaussian.

To see why this holds, note that each atom in the training set is deterministically labeled with $s \in \{0, 1\}$ via RDKit. Therefore, Term.(18) is equivalent to:

$$q(\mathcal{G}_{i,t} \mid \boldsymbol{\mathcal{G}}_0) = q(\mathcal{G}_{i,t} \mid s_{i,t} = 1, \boldsymbol{\mathcal{G}}_0)q(s_{i,t} = 1 \mid \boldsymbol{\mathcal{G}}_0) + q(\mathcal{G}_{i,t} \mid s_{i,t} = 0, \boldsymbol{\mathcal{G}}_0)q(s_{i,t} = 0 \mid \boldsymbol{\mathcal{G}}_0) \tag{21}$$

Similar to the proof of **Lemma 1** in the DDIM paper(Song et al., 2020), under the assumption of $q(\mathcal{G}_{i,t-1} \mid \boldsymbol{\mathcal{G}}_t, \boldsymbol{\mathcal{G}}_0)$ and $q(\mathcal{G}_{i,t} \mid \boldsymbol{\mathcal{G}}_0)$, we now aim to show that:

$$q(\mathcal{G}_{i,t-1} \mid \boldsymbol{\mathcal{G}}_0) = q(s_{i,t} = 1 \mid \boldsymbol{\mathcal{G}}_0)q(\mathcal{G}_{i,t-1} \mid s_{i,t} = 1, \boldsymbol{\mathcal{G}}_0) + q(s_{i,t} = 0 \mid \boldsymbol{\mathcal{G}}_0)q(\mathcal{G}_{i,t-1} \mid s_{i,t} = 0, \boldsymbol{\mathcal{G}}_0) \tag{22}$$

*Proof.* We begin with:

$$q(\mathcal{G}_{i,t-1} \mid \boldsymbol{\mathcal{G}}_0) = \int_{\boldsymbol{\mathcal{G}}_t} q(\boldsymbol{\mathcal{G}}_t \mid \boldsymbol{\mathcal{G}}_0)q(\mathcal{G}_{i,t-1} \mid \boldsymbol{\mathcal{G}}_t, \boldsymbol{\mathcal{G}}_0) \, d\boldsymbol{\mathcal{G}}_t \tag{23}$$

Using Bayes' rule:

$$q(s_{i,t} \mid \boldsymbol{\mathcal{G}}_t, \boldsymbol{\mathcal{G}}_0) = \frac{q(\boldsymbol{\mathcal{G}}_t \mid s_{i,t}, \boldsymbol{\mathcal{G}}_0)q(s_{i,t} \mid \boldsymbol{\mathcal{G}}_0)}{q(\boldsymbol{\mathcal{G}}_t \mid \boldsymbol{\mathcal{G}}_0)} \tag{24}$$

we substitute into the integral:

$$
\begin{aligned}
q(\mathcal{G}_{i,t-1}|\boldsymbol{\mathcal{G}}_0) &= \int_{\boldsymbol{\mathcal{G}}_t} q(\boldsymbol{\mathcal{G}}_t|\boldsymbol{\mathcal{G}}_0)\Big[q(\mathcal{G}_{i,t-1}|s_{i,t}=1,\boldsymbol{\mathcal{G}}_t,\boldsymbol{\mathcal{G}}_0)q(s_{i,t}=1|\boldsymbol{\mathcal{G}}_t,\boldsymbol{\mathcal{G}}_0) \\
&\qquad + q(\mathcal{G}_{i,t-1}|s_{i,t}=0,\boldsymbol{\mathcal{G}}_t,\boldsymbol{\mathcal{G}}_0)q(s_{i,t}=0|\boldsymbol{\mathcal{G}}_t,\boldsymbol{\mathcal{G}}_0)\Big]d\boldsymbol{\mathcal{G}}_t \\
&= \int_{\boldsymbol{\mathcal{G}}_t} q(\mathcal{G}_{i,t-1}|s_{i,t}=1,\boldsymbol{\mathcal{G}}_t,\boldsymbol{\mathcal{G}}_0)q(\boldsymbol{\mathcal{G}}_t|s_{i,t}=1,\boldsymbol{\mathcal{G}}_0)q(s_{i,t}=1|\boldsymbol{\mathcal{G}}_0)\, d\boldsymbol{\mathcal{G}}_t \\
&\qquad + \int_{\boldsymbol{\mathcal{G}}_t} q(\mathcal{G}_{i,t-1}|s_{i,t}=0,\boldsymbol{\mathcal{G}}_t,\boldsymbol{\mathcal{G}}_0)q(\boldsymbol{\mathcal{G}}_t|s_{i,t}=0,\boldsymbol{\mathcal{G}}_0)q(s_{i,t}=0|\boldsymbol{\mathcal{G}}_0)\, d\boldsymbol{\mathcal{G}}_t \\
&= q(s_{i,t}=1|\boldsymbol{\mathcal{G}}_0)q(\mathcal{G}_{i,t-1}|s_{i,t},\boldsymbol{\mathcal{G}}_0) + q(s_{i,t}=0|\boldsymbol{\mathcal{G}}_0)q(\mathcal{G}_{i,t-1}|s_{i,t}=0,\boldsymbol{\mathcal{G}}_0)
\end{aligned}
$$

$\square$

## A.4 DETAILS OF THE MODEL ARCHITECTURE

We follow the implementation of (Hoogeboom et al., 2022), where the EGNN is composed of equivariant convolutional layers, that is, $\mathbf{x}^{l+1}, \mathbf{h}^{l+1} = \text{EGCL}[\mathbf{x}^l, \mathbf{h}^l]$:

$$
\mathbf{m}_{ij} = \phi_e\left(\mathbf{h}_i^l, \mathbf{h}_j^l, d_{ij}^2, a_{ij}\right), \tag{25}
$$

$$
\tilde{e}_{ij} = \phi_{inf}(\mathbf{m}_{ij}), \tag{26}
$$

$$
\mathbf{h}_i^{l+1} = \phi_h\left(\mathbf{h}_i^l, \sum_{j\neq i}\tilde{e}_{ij}\mathbf{m}_{ij}\right), \tag{27}
$$

$$
\mathbf{x}_i^{l+1} = \mathbf{x}_i^l + \sum_{j\neq i}\frac{\mathbf{x}_i^l - \mathbf{x}_j^l}{d_{ij}+1}\phi_x\left(\mathbf{h}_i^l, \mathbf{h}_j^l, d_{ij}^2, a_{ij}\right), \tag{28}
$$

where $\phi_e, \phi_{inf}, \phi_h$ and $\phi_x$ are Multilayer Perceptrons, $d_{ij} = ||\mathbf{x}_i^l - \mathbf{x}_j^l||_2$.

**Property-guided Refinement.** We have introduced differentiated noise schedules and denoising strategies for scaffold and substituent atoms to better reflect their distinct structural roles. However, the asymmetric treatment can influence downstream molecular properties—such as HOMO (Highest Occupied Molecular Orbital) energy levels—which are known to be differently affected by scaffolds and substituents (Sicard et al., 2024; Góger et al., 2023). If the denoising process disproportionately perturbs one component, it may lead to inconsistencies in the electronic structure that deviate from realistic chemical behavior. To account for this, we incorporate a property-guided denoising gradient that steers the generation process toward molecules with desirable properties, thereby preserving structure–property consistency throughout the diffusion trajectory. Specifically, as illustrated in the red box of Fig. 1, we train a property predictor $p_\phi(c|\hat{\mathbf{s}}_t, \boldsymbol{\mathcal{G}}_t)$, which is trained to estimate a target molecular property $c$ based on the noisy molecule and the scaffold probability prediction. In this work, we focus on the HOMO energy as the target property, though the framework can be flexibly extended to other quantum or physicochemical properties. Following the classifier guidance strategy proposed in Dhariwal & Nichol (2021), we modify the reverse denoising update in Eq.(6) by incorporating an additional property-guidance term $\eta_1\rho_t\nabla_{\boldsymbol{\mathcal{G}}_t}\log p_\phi(c|\hat{\mathbf{s}}_t, \boldsymbol{\mathcal{G}}_t)$, where $\eta_1$ is a hyperparameter that controls the strength of the property guidance.

## A.5 DETAILS OF THE EXPERIMENTS

**Scaffold/substituent Label** We use RDKit to determine the scaffold/substituent label for each atom in the molecule. Specifically, we use the MurckoScaffold.GetScaffoldForMol method from RDKit to extract the scaffold atoms of a molecule, and label all remaining atoms as substituent. As we want to label every atom of the molecules with RDkit, all the molecules in QM9 dataset have to pass the validation check to generate the scaffold/sustituent label. Thus, the training/validation/test set will

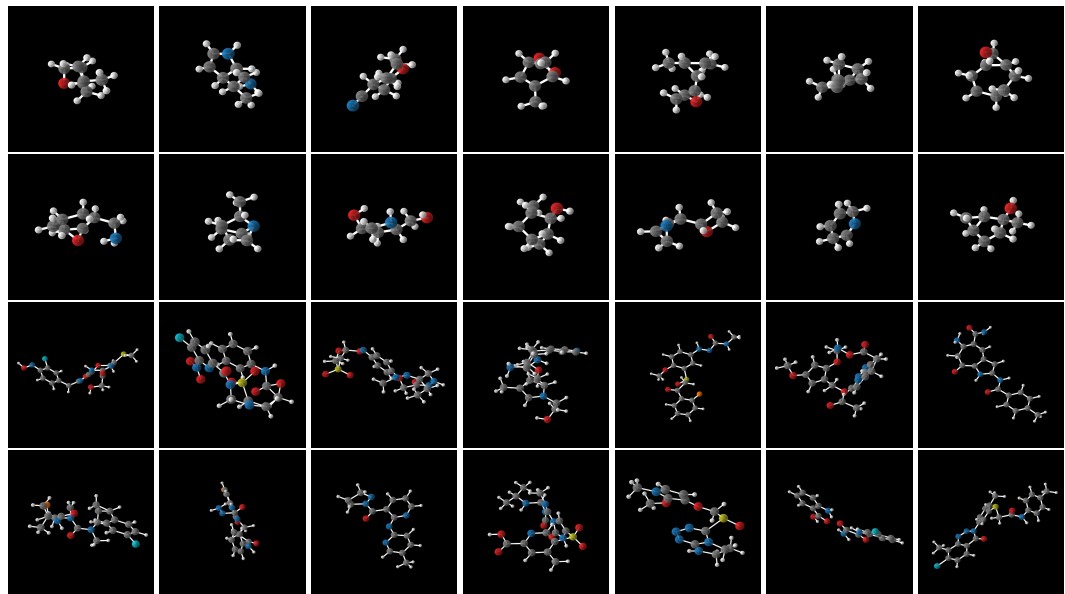

Figure 6: Samples generated by S$^2$-HDM trained on QM9 (top 2 rows) and GEOM-Drugs (bottom 2 rows).

not be identical as previous papers (Hoogeboom et al., 2022) (Xu et al., 2023). We split the whole dataset as training/validation/test set with 100k/6k/10k samples respectively.

**Hyperparameters** S$^2$-HDM has 256 hidden features and 15 layers for QM9 dataset, and 256 hidden features and 4 layers for GEOM-Drugs dataset. It is trained with batch size 64 and Adam optimizer with learning rate $10^{-4}$ on 1100 epochs. Diffusion step of the diffusion process is set to $T = 1000$. $\eta_0$ and $\eta_1$ are both set to 1.

**Compute Resource** We train our model on a single NVIDIA RTX A5000 for around 9 days.

More generated molecule samples are listed in Figure 6.

### A.6 DATASET DESCRIPTION

The QM9 dataset (Ramakrishnan et al., 2014) is a widely used benchmark that provides molecular properties and atomic coordinates for approximately 130,000 small molecules, each containing up to 9 heavy atoms (a total of 29 atoms, including hydrogens). Also, we test the proposed method on the GEOM-DRUG dataset (Axelrod & Gomez-Bombarelli, 2022). GEOM-DRUG dataset consist of complex organic compounds with a maximum of 181 atoms and average of 44.2 atoms across five distinct atomic species. This dataset encompasses ∼37 million conformations corresponding to ∼450,000 unique molecules, each annotated with energy levels and statistical weights.

### A.7 BROADER IMPACTS

Generative models possess powerful modeling and generalization capabilities; however, such capabilities also pose potential risks, as they may be exploited to generate molecules with unknown or harmful toxicity. If deployed in real-world industrial settings, it is crucial to conduct thorough analyses and experimental validation of the generated molecules to prevent adverse impacts on human health and the environment.

### A.8 USAGE OF LLMS

In this work, large language models (LLMs) are primarily employed as auxiliary tools to enhance the research workflow. Specifically, we leverage LLMs for two main purposes: (i) refining and polishing

the textual presentation to ensure clarity and readability; and (ii) assisting in the development of data visualization code, thereby streamlining the process of transforming experimental results into interpretable figures.

