# OpenReview forum: "Scaffold–Substituent Differentiated Diffusion for Hierarchical Molecule Generation"
_ICLR.cc/2026/Conference — ICLR 2026 Conference Withdrawn Submission_

### Official Review · Reviewer_BdJq · 2025-10-16

**Soundness:** 3
**Presentation:** 3
**Contribution:** 2
**Rating:** 2
**Confidence:** 4

**Summary:**

This paper introduces a hierarchical diffusion model designed to differentiate between scaffold and substituent atoms by employing two distinct noise schedulers and a role-classification module. The model trains a classifier to predict the role of each atom, enabling dynamic role estimation during sampling. The reverse diffusion process is then structured to prioritize scaffold formation in the early stages, followed by a flexible exploration of substituent atoms in later stages.

**Strengths:**

- The paper is well-written and well-structured.
- The authors motivate the problem of structure-aware diffusion process well, as prior studies work on an atom-level, while discarding the hierarchical structure of molecules.
- The authors validate the effectiveness of their proposed method, demonstrating improved performance on both unconditional and conditional generation tasks.

**Weaknesses:**

- To further support the claims made in the paper and underscore the relevance of the problem, it is important to evaluate how closely the distribution of generated scaffolds and functional groups aligns with that of the training data, and to compare these results with diffusion baselines that treat all atoms the same.
- The related work section is missing relevant studies on scaffold and substituent design. For example, DiffSBDD [1] performs inpainting for scaffold or fragment growth, and UniGuide [2] introduces guidance in a training-free manner during sampling.
- According to Tables 1 and 2, the results of S2-HDM include property refinement guidance, which may make the comparison with autoregressive (AR), flow-based, and diffusion baselines unfair. It would be helpful to include results without property refinement for a fairer comparison.
- The authors claim that their approach prioritizes scaffold generation followed by flexible substituent exploration. However, they do not report the diversity of the generated samples under either unconditional or conditional generation tasks.

**Questions:**

- Could you apply the trained classifier and the differentiated noise schedulers for scaffolds and functional groups during sampling with a given pretrained model (e.g., EDM) to assess whether this setup independently improves performance? Or does this correspond to the third row in the ablation study (Table 2)?
- Can you provide more details on the S2-HDM model architecture?
- Would it be possible to use the model for substructure conditioning tasks, such as scaffold hopping or scaffold elaboration [1,2], by using task-specific noise schedulers?

[1] Schneuing, Arne, et al. "Structure-based drug design with equivariant diffusion models." Nature Computational Science 4.12 (2024): 899-909.
[2] Ayadi, Sirine, et al. "Unified guidance for geometry-conditioned molecular generation." Advances in Neural Information Processing Systems 37 (2024): 138891-138924.

---

### Official Review · Reviewer_sfN9 · 2025-10-26

**Soundness:** 3
**Presentation:** 3
**Contribution:** 1
**Rating:** 2
**Confidence:** 3

**Summary:**

This paper proposes S$^2$-HDM (Scaffold-Substituent Hierarchical Diffusion Model), a diffusion-based generative model for 3D molecular generation that applies differentiated noise schedules to scaffold and substituent atoms.

**Strengths:**

1.  The paper draws clear inspiration from traditional drug design practices (scaffold hopping and lead optimization), which makes the hierarchical treatment of molecules quite intuitive and scientifically grounded.
2. The model shows consistent improvements across multiple metrics on standard benchmarks.

**Weaknesses:**

1. While the application is interesting, the technical contributions feel somewhat incremental. The main idea is essentially applying different noise schedules to different atom types. This is not particularly novel from a diffusion modeling perspective. The S$^2$ classifier is just a standard binary classifier added to the architecture. The property-guided refinement is borrowed directly from existing classifier guidance methods. Since that,  the claims are not well-supported, and the "research gap" framing feels a bit forced: existing methods don't necessarily fail to model hierarchy, they just approach it differently.
2. A significant limitation is that the model requires RDKit to generate scaffold/substituent labels during training. This means the "implicit learning" claim is somewhat overstated since you still rely on rule-based tools to define the hierarchy.
3. The dataset had to be filtered (100k/6k/10k split differs from baselines), making direct comparison slightly unfair
4. The paper claims the model learns hierarchy "implicitly" but actually uses explicit RDKit labels during training. What happens if these labels are noisy or incorrect?
5. No analysis on molecules where the scaffold/substituent decomposition is ambiguous or subjective
6. Some notation is inconsistent, and the proof in A.3 is quite verbose and could be streamlined
7. Why is the classifier only applied when t < 0.4T? This seems quite restrictive and is justified only by empirical observation rather than principled reasoning. Moreover, the model introduces several hyperparameters (η0, η1, ψt formula, 0.4T threshold). How sensitive is performance to these choices is not discussed.

**Questions:**

See weeknesses.

---

### Official Review · Reviewer_F5Ju · 2025-10-28

**Soundness:** 2
**Presentation:** 3
**Contribution:** 2
**Rating:** 4
**Confidence:** 5

**Summary:**

This model bakes the medicinal-chemistry idea of scaffold first, then decorate into the generative process by using different noise schedules for scaffold vs. substituent atoms—scaffolds are denoised early to set a stable core; substituents are explored later for diversity. During denoising, an EGNN predicts each atom’s role so the sampler can apply the right schedule—no predefined functional groups are required at inference, so the hierarchy is learned implicitly. (Training labels come from RDKit Murcko scaffolds. A frozen property predictor provides classifier-guidance gradients during mid/late timesteps

**Strengths:**

Unlike prior methods requiring: pre-defined functional groups, two-stage separate optimization ,and rule-based scaffold/substituent definitions this model learns the hierarchical structure implicitly through: An S² classifier that dynamically predicts atom roles during denoising and joint optimization of scaffold and substituent in a single framework. Evidence shows improvement over vanilla EDM model.

**Weaknesses:**

1. Hierarchical Noise Scheduling itself does not seem novel. differentiated noise schedules is not a new thing in molecule generation. For instance refer to MolDiff.

2. Some SOTA models available for comparison seem missing

**Questions:**

1. Can you add comparison with SLDM (Straight-Line Diffusion Model, 2025) and Geometry-Complete Diffusion for 3D Molecule
Generation and Optimization and MolDiff: Addressing the Atom-Bond Inconsistency Problem in 3D Molecule Diffusion Generation) ? Recent SOTA models seem to be missing. Especially the geom tasks and efficiency wise results.

2. Property-guided Refinement improves performance according to ablation. can this be used to other models for fair comparison?

3. Can the authors explain/compare with extra experiments related to geometric qualities (refer to MolDiff: Addressing the Atom-Bond Inconsistency Problem in 3D Molecule Diffusion Generation)

---

### Official Review · Reviewer_Ukvb · 2025-10-30

**Soundness:** 2
**Presentation:** 2
**Contribution:** 2
**Rating:** 4
**Confidence:** 4

**Summary:**

Through this paper, the authors propose Scaffold–Substituent Hierarchical Diffusion Model (S $^2$-HDM) to tackle the scaffold hopping and lead optimization problems with a single generative framework. Specifically, S$^2$-HDM introduces a differentiated noise schedule for scaffold and substituent atoms, enabling learning the scaffold and substituent hierarchy without pre-defined functional groups.

**Strengths:**

- The authors provided the codebase.
- The concept figure aids in understanding the work.
- The writing is easy to follow.

**Weaknesses:**

I will combine the *Weaknesses* section and the *Questions* section. My concerns are as follows:
- The framework relies on computational chemistry tools such as RDKit to identify the scaffold–substituent structure during training, which can be a critical error source. Moreover, during denoising, an auxiliary classifier (S$^2$ classifier) is used to identify the role of each atom given the noisy input, but there is no analysis on the prediction performance of the classifier.
- The main weakness of this paper is that the experimental design is insufficient. For example, quantitative estimate of druglikeness (QED), retrosynthetic accessibility (RA), medicinal chemistry filter (MCF), and synthetic accessibility (SA) are important metrics that evaluate chemical quality of generated molecules. I highly recommend to report these values following the HierDiff paper [1]. The coverage and matching metrics for molecular conformation quality evaluation also need to be reported, following the HierDiff paper [1].

---

**References:**

[1] Qiang et al., Coarse-to-fine: a hierarchical diffusion model for molecule generation in 3d, ICML, 2023.

**Questions:**

Please see the *Weaknesses* section for my main concerns.

---

### Note · Authors · 2026-01-27

I have read and agree with the venue's withdrawal policy on behalf of myself and my co-authors.

---

### Meta-Review · Area_Chair_G5o6 · 2026-01-07

**Summary:**

Reviewers raised several valid concerns as detailed below. However, authors did not attempt to address any of the concerns.

**Reviewer Concerns:**

None of the concerns were addressed as no rebuttal was presented.

**Reviewer Scores:**

None.

---

### Decision · Program_Chairs · 2026-01-26

Reject